# ACCELERATE HIGH-QUALITY DIFFUSION MODELS WITH INNER LOOP FEEDBACK

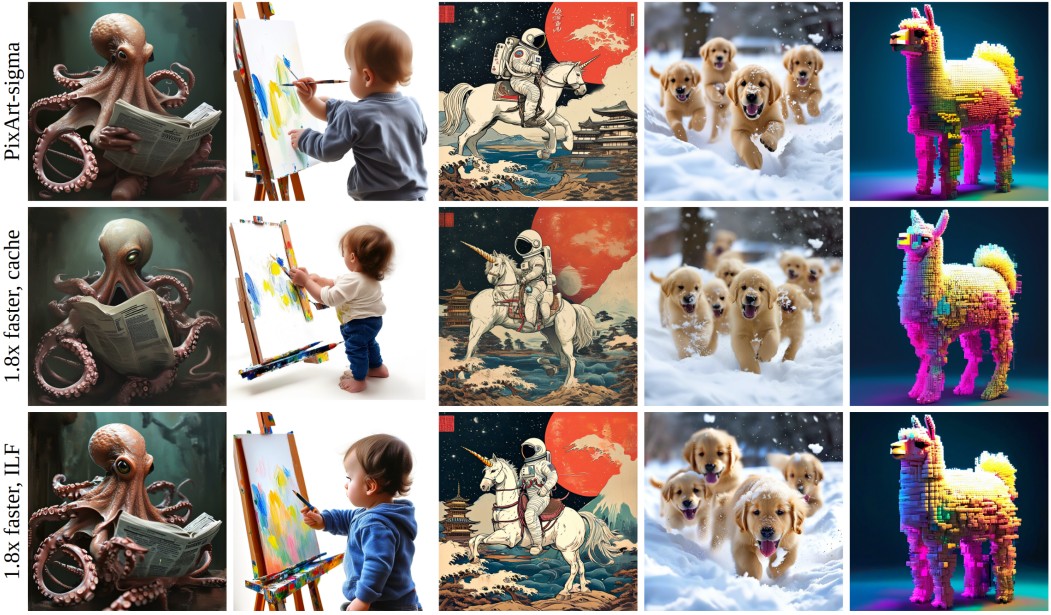

Figure 1: PixArt-sigma 1024x1024 images, generated with 20 steps using DPM-Solver++ (top) vs. PixArt-sigma with caching (middle) vs. PixArt-sigma with **ILF** (bottom). ILF produces high quality images **1.8x** faster, measured on H100 GPUs.

## ABSTRACT

We propose **I**nner **L**oop **F**eedback (**ILF**), a novel approach to accelerate diffusion models' inference. ILF trains a lightweight module to predict future features in the denoising process by leveraging the outputs from a chosen diffusion backbone block at a given time step. This approach exploits two key intuitions; (1) the outputs of a given block at adjacent time steps are similar, and (2) performing partial computations for a step imposes a lower burden on the model than skipping the step entirely. Our method is highly flexible, since we find that the feedback module itself can simply be a block from the diffusion backbone, with all settings copied. Its influence on the diffusion forward can be tempered with a learnable scaling factor from zero initialization. We train this module using distillation losses; however, unlike some prior work where a full diffusion backbone serves as the student, our model freezes the backbone, training only the feedback module. While many efforts to optimize diffusion models focus on achieving acceptable image quality in extremely few steps (1-4 steps), our emphasis is on matching best case results (typically achieved in 20 steps) while significantly reducing runtime. ILF achieves this balance effectively, demonstrating strong performance for both class-to-image generation with diffusion transformer (DiT) and text-to-image generation with DiT-based PixArt-alpha and PixArt-sigma. The quality of ILF's 1.7x-1.8x speedups are confirmed by FID, CLIP score, CLIP Image Quality Assessment, ImageReward, and qualitative comparisons.

Figure 2: ILF uses a lightweight, learnable feedback module to create a powerful inner loop within a diffusion model. Instead of computing a forward through all backbone blocks, in order, we choose some block, feed its output features to the feedback, then feed those features back to some earlier blocks in the model, modified by a learnable scaling term. The feedback's objective is essentially to predict features corresponding to some future diffusion time step, so the resulting noise prediction is more reliable for the model's current step.

# 1 INTRODUCTION

Since its introduction as an alternative to generative adversarial networks (GANs) Goodfellow et al. (2014) for image synthesis Dhariwal & Nichol (2021), diffusion has been one of the most prominent methods for generative tasks. These methods deliver stable training, high quality generations, and easy alignment to a variety of conditions for generative tasks Yang et al. (2024). However, the actual generation process is quite expensive. While GANs generate images in a single model forward pass, diffusion models require many model forward passes to iteratively progress from random noise to clean images. As a result, many researchers have focused on trying to improve the efficiency of diffusion models, while retaining the quality. Some of these are training-free, focusing on caching features for cheaper inference; others involve expensive distillation to dedicated-purpose few step models.

We propose inner loop feedback (ILF) for diffusion models, seeking to achieve higher quality at better efficiency than caching, as shown in Figure 1, without the training cost and inflexibility of distillation-based approaches. With this approach, we can take any frozen pre-trained transformer-based diffusion model, and make each of its steps more powerful by training a new block, the feedback module, to take features from a block $b$ at one step, $t$, and predict features for prior blocks $b - 1, b - 2, ..., b - l$ corresponding to the next step, $t - r$, as shown in Figure 2. In contrast to these distillation-based methods that seek to learn models that can achieve *reasonable* quality with 4 or fewer inference steps, we focus instead on matching or surpassing the *best-case* performance of the original model, in less time. Furthermore, with our approach, one does not need to store an entire additional set of models weights, instead only those corresponding to the lightweight learnable module.

From caching literature focused on the U-Net architecture, we already know that features for these U-Net diffusion models are very similar for a given block, $b$, at adjacent time steps, $t$ and $t - r$ Ma et al. (2023); Wimbauer et al. (2024). We find this remains consistent for transformer-based diffusion models, in Figure 3, with an implementation described in Section 2.2. Caching methods leverage this concept to skip computation for certain blocks at certain steps, simply re-using the prior features. However, as Figure 3 also shows, leveraging these similar adjacent features changes the make-up of the features themselves.

The caching schedule itself is noticeable; since features are cached on every other step, instead of smooth change over blocks and time, feature pairs become noticeable on the time axis. Furthermore, except for the last steps, the final model outputs become less distinct between neighboring steps. The same phenomenon manifests when we consider how the features evolve across blocks (Figure 4); once we introduce caching, features of the last block do not become as dissimilar to the first step for the steps on which the caching occurs. This is clearly not optimal, and results in poorer quality images, as shown in Figure 5. Caching often loses key details, and produces blurrier, less appealing

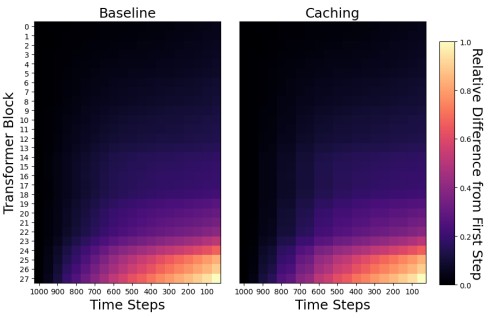 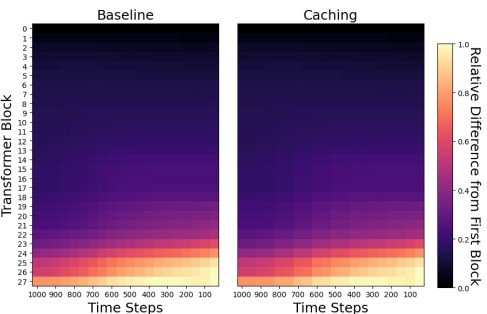

Figure 3: Change in features across time steps, measured for each block as difference from that block's feature at $t = 1000$, normalized by dividing by the maximum difference across both plots. Caching reduces the degree to which the features change over time.

Figure 4: Change in features across blocks, measured for each block at each time step as difference from the first block's feature at that time step, normalized as in Figure 3. The trend with caching is similar here as when measuring difference over time.

generations. This motivates our inner loop feedback; we want to take advantage of the redundancy between blocks across time, without compromising the quality of the final images.

For flexibility of adaptation to novel architectures, we propose leveraging the model architecture itself to design the feedback module. The module consists of a single block, copied from the block design of the corresponding pre-trained diffusion transformer backbone. That is, for the basic DiT, our feedback module is a single DiT block. This way, our feedback handles conditions, input sizes, and output sizes in the same way as the base model.

Different diffusion models have different learned weight values, and training quickly, without over-fitting, is quite challenging. To allow for fast training we use distillation. However, unlike prior works, we do not have a separate teacher and student. Instead, the model without feedback is the teacher, and the model with feedback is the student – but only the feedback is learnable. Furthermore, while prior works perform multiple teacher iterations, which is expensive, we propose an approach that allows us to only use one. We find this Fast Approximate Distillation works equally well, at lower cost. To avoid over-fitting, we add Learnable Feedback Rescaling, where we learn an integer scaling term on the feedback before we add it to the features in the pretrained model. Initializing this to zero allows the model to learn quickly from the distillation, without diverging due to excessively large error. Neither Fast Approximate Distillation nor Learnable Feedback Rescaling require any hand-tuned hyperparameters, allowing for easy implementation and extension to other models and data.

One challenge introduced by our approach is that since we attempt to predict features corresponding to future steps, it is not immediately clear how to set the noise steps for the backbone and scheduler. While we find that using default schedulers works well, this does not hold true for the model backbone itself, where we must use a rescaled step for the time conditioning at inference time. Furthermore, we find that for larger inner loops, it is often best to skip feedback on some steps altogether. We instantiate these findings collectively as Feedback-aware Inference Scheduling.

In summary, we achieve high quality efficient generation with the following contributions:

- We propose diffusion with Inner Loop Feedback (ILF), a feedback mechanism which creates a powerful inner loop within transformer-based diffusion backbones for optimal time-quality trade-offs.

- We develop Learnable Feedback Rescaling and Fast Approximate Distillation for speedy training, and Feedback-aware Inference Scheduling to adaptively leverage the power and speed of ILF at inference time.

- We achieve superior results to caching for 1.7x-1.8 speedups compared with the baseline model, with an average +7.9 improvement for Image Reward, +0.14 CLIP, -0.22 MJHQ FID, and +1.4 CLIP IQA Score.

## 2 RELATED WORKS

### 2.1 DIFFUSION FOR IMAGE GENERATION

**Diffusion Fundamentals.** Diffusion models Ho et al. (2020); Nichol & Dhariwal (2021); Dhariwal & Nichol (2021) consider a forward noising process. Given some distribution $q(x_0)$, a sample $x_0$ is noised in steps accordingly to a schedule, $\{\beta_t\}_{t=1}^T$, where at any time step $t$, we can calculate the noised sample, $x_t$, as

$$x_t = \sqrt{\bar{\alpha}_t}x_0 + \sqrt{1 - \bar{\alpha}_t}\epsilon, \epsilon \sim \mathcal{N}(0, \mathbf{I}) \tag{1}$$

with $\alpha_t := 1 - \beta_t$ and $\bar{\alpha}_t := \prod_{i=0}^t \alpha_i$. The diffusion model is instantiated as a neural network that reverses the forward noising process, by predicting $\epsilon_t$ that ought to be removed from $x_t$ to predict $x_{t-1}$.

**Model Architecture.** Early diffusion models rely on U-Net architectures Ronneberger et al. (2015); Ho et al. (2020); Dhariwal & Nichol (2021), processing images (or noised images) into features in an encoder, then back to images (or noise, in pixel space) with a decoder, with connections between symmetric encoder and decoder blocks. For the sake of efficiency, subsequent methods perform diffusion on latent representations Rombach et al. (2022) from pre-trained variational auto-encoders Kingma & Welling (2014). Originally proposed for image generation, these models are also well-suited for image editing Kawar et al. (2023); Brooks et al. (2023) and video generation Ho et al. (2022); Blattmann et al. (2023); Liu et al. (2024b). These models can be conditioned on text encodings Saharia et al. (2022); Ramesh et al. (2022); Nichol et al. (2022); Ruiz et al. (2023); Podell et al. (2023) from powerful models including CLIP Radford et al. (2021) and T5 Raffel et al. (2023). Recently, to allow for more flexible scaling, transformer-based diffusion models Peebles & Xie (2023); Bao et al. (2023) have become the predominant architecture for state-of-the-art diffusion models Chen et al. (2023; 2024b;a); Esser et al. (2024); Liu et al. (2024b). Due to the recent trend towards diffusion transformers, we choose to focus our work primarily on this family of architectures, including the original DiT Peebles & Xie (2023) for class-to-image generation, and PixArt-alpha Chen et al. (2023) and PixArt-sigma Chen et al. (2024a) for text-to-image generation.

**Inference Scheduling.** While diffusion models are typically trained on 1000-step schedules, inference is performed at much lower steps, with schedulers to handle timestep spacing, as well as forward and reverse noising Song et al. (2022); Karras et al. (2022); Lu et al. (2023). Recent work investigates non-uniform, model-specific timestep spacing Sabour et al. (2024). We propose substantial inference-time improvements for ILF in Section 3.3, but these are all compatible with existing inference schedulers.

### 2.2 FASTER DIFFUSION INFERENCE

A significant body of work focuses on speeding up diffusion inference to generate good images in fewer steps. The majority of these approaches can be divided between training-free caching approaches, and training-based distillation approaches.

**Caching.** Prior caching work focuses mainly on U-Net architectures. While the methods share a key intuition, that features are similar at adjacent time steps, their configurations differ. Some cache only encoder features Li et al. (2023), others cache outer layers of both encoder and decoder Ma et al. (2023), and others automatically discover ways to cache a variety of layers Wimbauer et al. (2024). Concurrent work has started to approach caching for DiTs Selvaraju et al. (2024); Liu et al. (2024a), including some work which focuses on learnable routing for the caching Ma et al. (2024). One major difference between U-Nets and DiTs is the absence of encoder-decoder distinction, which changes the caching approach substantially. Furthermore, many of these approaches focus on generation of lower resolution images, using class conditioning, with many time steps. By contrast, more modern text-to-image models use fewer steps. With fewer steps, the feature maps change much less smoothly over time, mitigating the suitability of caching; furthermore, errors and blurriness become even more glaring at higher resolution (see Figure 5 for examples). Nevertheless, we implement caching as a point of comparison for ILF.

For this caching, we store the attention and feedforward results for each cacheable block on the first step. Then, at subsequent steps, for all cacheable blocks, we only recompute attentions and

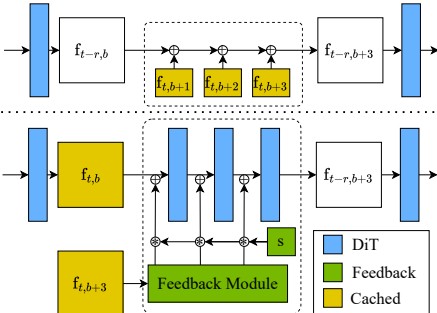

Figure 5: We compare typical 20 step diffusion inference to caching for PixArt-alpha, 512x512 images. We cache the middle 14 blocks, re-computing features every other step. Caching, while more efficient, sometimes results in quality degradation – loss of detail (no faces in left-most image), less appealing design (middle images), and blurriness (rightmost image, zoom in on eyes, ears, hair, and mouth).

Figure 6: Caching (top) vs. ILF (bottom). We show how we use a partial diffusion forward pass to compute $f_{t,b+3}$, which we then use to compute $f_{t-r,b+3}$. We can use fewer of ILF's heavy steps to ultimately achieve better quality-time trade-offs than caching's cheap steps.

feedforwards on every other step; otherwise, we simply add the stored results to the new input hidden states. We illustrate this approach in Figure 6, and compare it to ILF. Unlike caching approaches, we train ILF with lightweight external module that increases the complexity of each forward pass, which allows us to achieve much better quality-speed trade-offs at inference time.

**Training or Finetuning.** Some approaches learn lightweight modules for predicting skip connections Jiang et al. (2023) or predicting steps based on prompt complexity Zhang et al. (2023). The majority of the literature tends to focus on knowledge distillation Hinton et al. (2015), progressive distillation Salimans & Ho (2022), guidance distillation Meng et al. (2023), and consistency distillation Luo et al. (2023). Unlike the majority of these works, we do not focus on generating images of *acceptable* quality in extremely low steps Lee et al. (2024); Sauer et al. (2023); Kohler et al. (2024); Yin et al. (2023); Xu et al. (2023b); instead, we seek to synthesize *maximum* quality images in the fewest possible steps.

## 3 APPROACH

### 3.1 INNER LOOP FEEDBACK DESIGN

We propose a lightweight learnable module that leverages similar intuitions to caching, but with a different mechanism and superior results. This method, illustrated in Figure 2, starts with some pre-trained, transformer-based diffuion model. Standard diffusion forward passes attempt to predict $\hat{\epsilon}(x_t, t)$, for some noised latent $x_t$ and time step $t$. By contrast, with our feedback mechanism, we attempt to make the forward pass more powerful, where we instead predict $\hat{\epsilon}(x_{t-r}, t-r)$, where $r$ is some positive integer, meaning $t - r$ is some subsequent time step. This allows us to generate high quality images with fewer, but more powerful, inference steps.

We design the feedback module itself by simply copying the architecture of the model blocks themselves, such that for a standard $N$ block DiT, we introduce a $(N + 1)^{\text{th}}$ block. However, instead of simply appending, prepending, or inserting the block, we dramatically alter the flow of information. We first set a location for the inner loop, denoted by the beginning ($b$) block, $B_b$, and the ending ($e$) block $B_e$. For some time step $t$, the feedback module takes as its input, the output of $B_e$, $f_{e,t}$, along with the embedded time and text conditions. The feedback model gives its output, $f_{\text{feed}}$. We then rescale the $f_{\text{feed}}$ for separately for each block in the inner loop, $\{B_b, B_{b+1}, ..., B_e\}$, by multiplying each by its corresponding learnable floating point scaling factor, $\{s_b, s_{b+1}, ..., s_e\}$. For the first block, $B_b$, we compute its result as

$$f_{b,t-r} = f_{\text{feed}} * s_b + f_{b-1,t} \tag{2}$$

We compute the features outputs of any subsequent block, $f_{i,t-r}$ for block $B_i$, with

$$f_{i,t-r} = f_{\text{feed}} * s_i + f_{i-1,t-r} \tag{3}$$

## 3.2 Training Inner Loop Feedback

One cannot train this feedback mechanism with basic random initialization; the magnitude of the feedback will be too large, and the training will diverge. Furthermore, standard training is needlessly slow. To keep the training stable and time-efficient, we leverage both novel Learnable Feedback Rescaling as well as Fast Approximate Distillation.

**Learnable Feedback Rescaling.** As mentioned in Section 3.1, we rescale the feedback outputs, $f_{\text{feed}}$ with some learnable scalar $s_i$ for each block in the inner loop. This simple multiplication operation is cheap, and allows us to use a single feedback computation to improve the features used by all blocks within the inner loop. Furthermore, by zero-initializing and learning $\mathbf{s}$ we are able to avoid needing to set any hard-coded hyperparameters.

**Fast Approximate Distillation** Standard diffusion models train with a reconstruction loss. We use this same loss, and a novel pseudo-self-distillation loss between the output of diffusion with ILF (student) and diffusion without ILF (teacher). To align with our objective to predict future noise outputs, we perform the distillation using less noisy images. Specifically, whenever our ILF input during training is noised to step $t$, we noise the teacher input to step $t/2$. We then compute standard mean squared error loss between their predictions. Note that while we refer to diffusion with ILF as the student, only the feedback module and the rescaling parameters are learnable. This novel formulation saves on training cost.

## 3.3 Feedback-aware Inference Scheduling

With ILF, we are training the feedback to take inputs for one time step $t$, and produce predictions for a future time step $t - r$. However, in practice, we still need to denoise the actual original input, $x_{\text{t}}$. Treating this as if it were a more clean input, $x_{\text{t-r}}$, and removing the corresponding amount of noise, would be counterproductive. So, we still use the sigmas corresponding to $t$ rather than $t - r$ for the backwards diffusion process itself. Thus, our method is akin to conditioning the model to generate a more reliable noise prediction, which can be used reliably for more spacing diffusion (fewer inference steps).

However, time step is not only used for the noise subtraction process. Rather, it is also a condition for the diffusion backbone itself. Hence, we must change the time condition used for all computation in the each forward pass that occurs *after* the feedback module forward. For the subsequent computation, we find that using an intermediate step, weighted for the size of the inner loop, is appropriate. So, for an inner loop with $m$ blocks in a model comprised of $n$ total blocks, and a scheduler with consecutive time steps $t$ and $t - i$, we compute the intermediate post-feedback time step, $t_{\text{post}}$, with

$$t_{\text{post}} = t - i * (n/m) \tag{4}$$

We refer to this strategy in Section 4 as "Rescaling," as opposed to "Uniform" computation of $t_{\text{post}} = t - i/2$. We also observe that as we continue to train the feedback mechanism, it will "overfit" – providing cluttered, over-saturated outputs. While on face this seems problematic, we actually find we can take advantage of it. First, we modify our rescaling to anneal over time. For $t = 999$, we use $t_{\text{post}}$ as in Equation 4. However, for subsequent steps, we instead compute $t_{\text{post}}$ as

$$t_{\text{post}} = t - \max(i * (n/m) * (t/1000), 10) \tag{5}$$

While we find this "Annealing" helps to improve results (see Figure 8), it is not fully optimal. By "Skipping" some feedback on some of the middle inference steps, we are able to perform inference even faster, while improving the quality from ILF. We provide some useful configurations and empirical exploration in Section 4.

## 4 Experiments

### 4.1 Experimental Setup

We use 5 pre-trained diffusion models, DiT, PixArt-alpha 512x512, PixArt-alpha 1024x1024, PixArt-sigma 512x512, and PixArt-sigma 1024x1024. All experiments and results are computed

Table 1: Main results, high-quality text-to-image generation, speedups compared to 20 step DPM-Solver++ generations. We bold the best efficient results; that is, the higher of each metric between ILF, caching, and the baseline at 12 steps. ILF is sometimes even better than the 20 step baseline.

| Settings | | | Latency | | Prompt-aware Metrics | | FID ↓ | CLIP Image Quality Assessment | | | |
|---|---|---|---|---|---|---|---|---|---|---|---|
| Model | Res. | # Steps | # Blocks | s / img | Image Reward | CLIP | MJHQ | Good | Noisy ↓ | Colorful | Natural |
| PixArt-alpha | 1024 | 20 | 560 | | 6.38 | 94.43 | 28.96 | 6.51 | 92.71 | 23.92 | 57.79 | 66.26 |
| PixArt-alpha | 1024 | 12 | 336 | 3.69 (1.7x) | 90.41 | 28.94 | 6.86 | **92.75** | 25.38 | 56.07 | 64.71 |
| PixArt-alpha w/ cache | 1024 | 20 | 326 | 3.63 (1.8x) | 82.49 | 28.86 | 6.85 | 91.20 | 29.47 | 50.38 | 63.38 |
| PixArt-alpha w/ ours | 1024 | 10 | 332 | 3.63 (1.8x) | **91.71** | **28.98** | **6.13** | 90.60 | **24.91** | **59.18** | **66.97** |
| PixArt-sigma | 1024 | 20 | 560 | 6.63 | 83.87 | 29.28 | 7.28 | 90.32 | 27.98 | 59.60 | 69.12 |
| PixArt-sigma | 1024 | 12 | 336 | 3.81 (1.7x) | **81.82** | 29.43 | 6.86 | **89.65** | 31.78 | 63.26 | 65.01 |
| PixArt-sigma w/ cache | 1024 | 20 | 326 | 3.75 (1.8x) | 71.93 | 29.33 | 7.44 | 84.24 | 38.49 | 48.02 | **72.24** |
| PixArt-sigma w/ ours | 1024 | 10 | 332 | 3.75 (1.8x) | 79.74 | **29.45** | **6.79** | 88.26 | **30.22** | **69.28** | 63.56 |
| PixArt-alpha | 512 | 20 | 560 | 1.06 | 92.03 | 29.06 | 7.13 | 92.79 | 17.17 | 66.17 | 51.59 |
| PixArt-alpha | 512 | 12 | 336 | 0.62 (1.7x) | 88.42 | 29.02 | 7.86 | **94.49** | 18.95 | **71.57** | 48.06 |
| PixArt-alpha w/ cache | 512 | 20 | 326 | 0.59 (1.8x) | 82.95 | 28.93 | **6.56** | 92.04 | 19.52 | 61.99 | 48.67 |
| PixArt-alpha w/ ours | 512 | 10 | 332 | 0.59 (1.8x) | **89.47** | **29.11** | 7.20 | 92.67 | **16.89** | 69.31 | **50.18** |
| PixArt-sigma | 512 | 20 | 560 | 1.14 | 94.17 | 29.12 | 7.99 | 89.57 | 20.04 | 65.67 | 52.69 |
| PixArt-sigma | 512 | 12 | 336 | 0.66 (1.7x) | 94.17 | 29.20 | 7.21 | **90.82** | **19.75** | 68.47 | 48.93 |
| PixArt-sigma w/ cache | 512 | 20 | 326 | 0.66 (1.7x) | 87.08 | 29.09 | 7.05 | 87.73 | 22.32 | 59.38 | **53.26** |
| PixArt-sigma w/ ours | 512 | 10 | 332 | 0.66 (1.7x) | **95.28** | **29.24** | **6.92** | 89.35 | 19.91 | **73.06** | 45.87 |

on NVIDIA H100 GPUs, unless otherwise specified, and scale up the quantity as necessary for each experiment. Whenever we train our feedback module for text-to-image, we use learning rate $10^{-6}$, batch size 2048, and train for 5 epochs across a proprietary set of 2 million high-quality text-image pairs. For class-to-image, we use learning rate $5 * 10^{-6}$, batch size 8192, and train for 10 epochs on the approximately 1,281,167 ImageNet Russakovsky et al. (2015) class-image pairs. Unless otherwise indicated, we use DPM-Solver++ Lu et al. (2023). For the base 28-block DiT with 749M frozen parameters, our feedback adds 26.7M learnable parameters. ILF adds 21.3M learnable parameters to 611M frozen parameters for both 28-block PixArt-alpha and 28-block PixArt-sigma.

To assess our performance, we rely on both examples and metrics. Unless otherwise specified, example images are drawn from sample prompts we provide in the supplementary material. For quantitative results, we compute Image Reward Xu et al. (2023a), using the prompts and procedure from the official code repository. We also compute MJHQ Li et al. (2024) FID with clean-fid Parmar et al. (2022), CLIP score Hessel et al. (2022) on the generations from complex prompts we provide in the supplementary, and CLIP IQA Wang et al. (2023) on images generated from the Image Reward prompts. When computing CLIP IQA, we report the standard CLIP IQA Score as "Good" (since it is the result of competing "Good" and "Bad" text prompts), as well as its measurements of "Noisy," "Colorful," and "Natural." In general we prioritize Image Reward due to its good correlation with human judgments, but other metrics offer further confirmation of our method's utility.

## 4.2 MAIN RESULTS

We show that our method works exceptionally well for fast, high quality text-to-image generation in Table 1. For settings for our method, we train feedback to create an inner loop from block $b = 8$ to block $b = 19$, and at inference we perform feedback only for the first two and last two steps. We outperform the caching baseline (where we cache the middle 18 blocks, re-computing features once every 3 steps) for nearly every metric across both models at both resolutions. Furthermore, we even achieve comparable or better results in many metrics compared to the inefficient baseline.

In addition to seconds per image, we measure latency by number of block forwards to generate the image. To compute block forwards, we add up the total number of passes through a transformer block. Since the blocks are all the same size and shape (including our feedback block), this is a straightforward, reliable way to compare complexity across methods.

While we sample a variety of metrics for thoroughness, none correlate perfectly with human judgment of quality. So, for further results and ablations, we show actual generated images in Figure 7. While Table 1 shows ILF performs well in the 1.8x speedup setting, here we show our results are clearly superior to caching even for less dramatic speedups, where our 1.5x has better visual quality than caching at 1.4x. For this setting, we use 12 steps instead of 10, and skip feedback for the inner 8 steps. Notice how even when it deviates from the content of the baseline, ILF provides more clear,

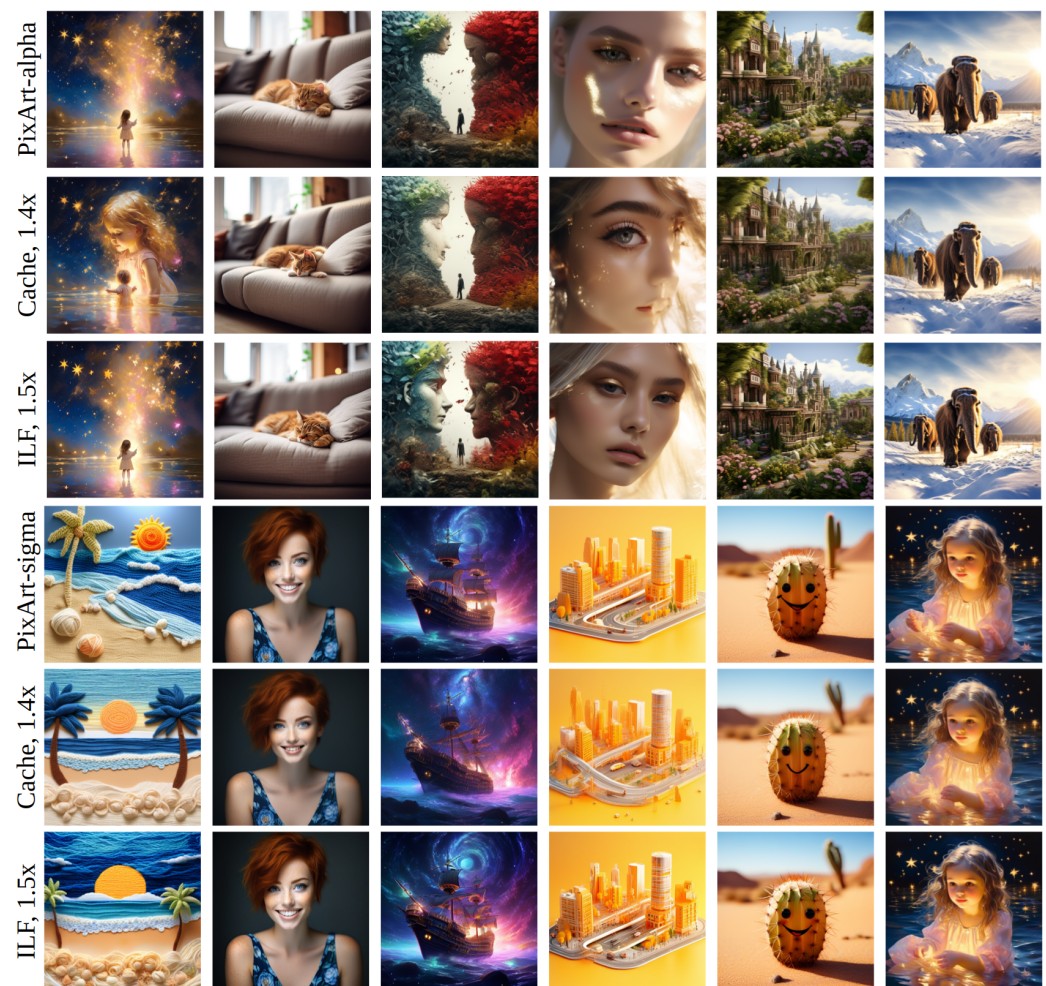

Figure 7: 512x512 results, alpha (top 3 rows) and sigma (bottom 3 rows), with baseline, caching, and our results, respectively. ILF yields images of similar content and quality to the un-accelerated baseline, and clearly superior to the caching, for both models. Zoom in for finer details.

appealing, detailed generations compared to the caching approach. See Figure 12 in the appendix for results from the MJHQ prompts, showing further evidence of ILF's good quality, at 1.8x speed.

## 4.3 ABLATIONS

We first show ILF works for class-to-image generation in Table 2. While the speedups are less dramatic, the ImageNet FID improvements are non-trivial. Since the method itself is designed primarily for text-to-image generation, we use this to showcase the flexibility of the method for a different task. We choose settings to safely give both some speedup and FID improvement, but with more tuning, or else aiming for equal FID, one could achieve better ImageNet FID with ILF.

We verify that our method is not overly sensitive to the location of the inner loop in Table 3. Indeed, as long as the loop is not at the end of the model, results are quite comparable among various settings. Note that for the smaller loops we only rescale the feedback, whereas for the larger loops we both rescale and skip feedback for the middle 8 steps.

We provide some understanding of the impact of training time on quality in Figure 8 and of the relationship between inference steps and image reward in Figure 9. Image quality increases over training time until it saturates around 5,000 iterations. However, not all inference strategies are equally well-suited. Similarly, quality increases with more inference steps. As Figure 9 suggests,

Table 2: Class-to-Image results, ImageNet, DiT 256x256. ILF consistently has better FID at better speed. While we choose settings here that consistently *outperform* the non-accelerated baseline FID, one could instead prioritize speed with ILF to *match*, rather than beat, the baseline.

| Model | # Steps | # Blocks | s / img | FID ↓ |
|---|---|---|---|---|
| DiT | 12 | 336 | 0.14 | 4.50 |
| DiT w/ ours | 10 | 304 | 0.13 (1.1x) | 4.06 (-0.47) |
| DiT | 25 | 700 | 0.29 | 3.96 |
| DiT w/ ours | 20 | 584 | 0.24 (1.2x) | 3.59 (-0.37) |
| DiT | 50 | 1400 | 0.57 | 3.56 |
| DiT w/ ours | 40 | 1144 | 0.47 (1.2x) | 3.31 (-0.25) |

Table 3: PixArt-alpha 512x512 loop size and location ablation, 12 steps. We compare small loops 3 different locations, to large loops at similar locations. We use our skipping inference scheduling for the larger loops to preserve the quality, which also gives better speedups.

| Loop Size | | Latency | | Metrics | |
|---|---|---|---|---|---|
| Start | End | # Blocks | s / img | Image Reward | MJHQ FID ↓ |
| 0 | 5 | 420 | 0.78 (1.36x) | **94.26** | 7.32 |
| 11 | 16 | 420 | 0.78 (1.36x) | 93.80 | 7.07 |
| 22 | 27 | 420 | 0.78 (1.36x) | 88.10 | 8.39 |
| 0 | 11 | 388 | 0.73 (1.44x) | 93.10 | **6.47** |
| 8 | 19 | 388 | 0.73 (1.44x) | 93.14 | 6.75 |
| 16 | 27 | 388 | 0.73 (1.44x) | 90.20 | 7.56 |

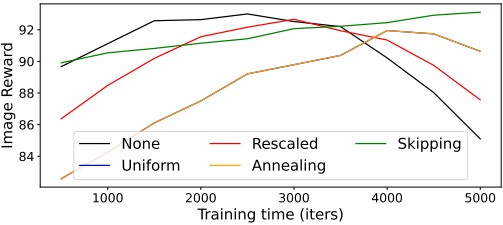

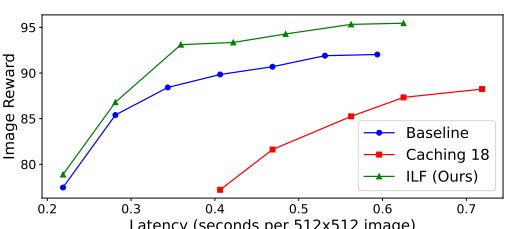

Figure 8: Training time effects on feedback scheduling, measured by Image Reward for PixArt-alpha 512x512.

Figure 9: Image Reward vs. inference time, PixArt-alpha 512x512 for baseline, caching, and ILF. Ours is superior.

our method has a substantial edge in quality across the range of intermediate to high steps (we neither consider nor report extremely low step results). As a disclaimer, Image Reward, while it correlates with human judgment better than most metrics, is still not perfect; from our observation, it is not overly sensitive to some of the lighting, sharpening, and over-detailing artifacts our method will introduce if its over-fitting is not properly mitigated.

To determine which steps for which to skip feedback, we perform an ablation, with sample generations shown in Figure 10. We find that skipping feedback for the inner steps yields the most consistently good results, which lines up with our intuition that the first steps are the most important for determining good layouts, and the last steps are quite important for guaranteeing good fine details. So, naturally, it is best to perform our powerful diffusion feedback on those steps.

We also demonstrate that training with our Fast Approximate Distillation is better than training without using distillation, in Figure 11. Furthermore, our results match results from training with the more expensive standard distillation (multiple teacher steps, in this case 8). Since instead of 8 teacher steps, we only need 1, we are able to achieve good results with cheaper training. For further ablations, exploration, and examples, see the Appendix.

## 5 CONCLUSION

We propose diffusion with Inner Loop Feedback (ILF), which lets us perform diffusion inference with fewer, more powerful inference steps. As a result, we can leverage pre-trained diffusion models to generate high-quality images in less time. With Learnable Feedback Rescaling and Fast Approximate Distillation, we are able to train feedback for efficient megapixel image generation in approximately 100 GPU hours. Our method outperforms the training-free caching baseline, and is substantially cheaper and more flexible than any distillation-based alternative. Future work could explore our method as a way to cheaply finetune a diffusion model on new data, as well as try to achieve better performance at extremely low steps. Additionally, with some adjustments (accounting for encoder-decoder skip connections), our work could be adapted for U-Nets, though we consider this out of scope due to the rising popularity of DiTs.

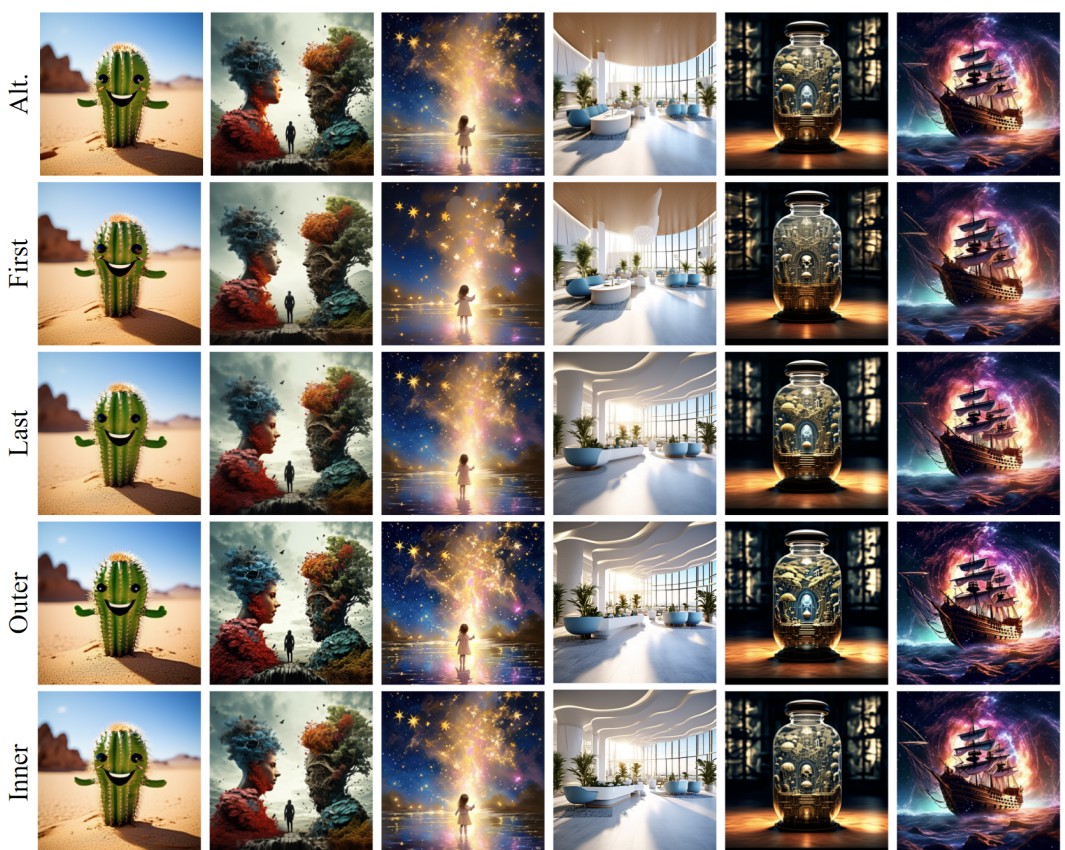

Figure 10: Visual examples of different skipping feedback for different time steps at inference time, skipping feedback for the alternating steps (top), first steps only (second), last steps only (third), outer steps (fourth) and then inner steps (bottom). Skipping feedback for inner steps is best, with good overall structure and high quality details, without distortions.

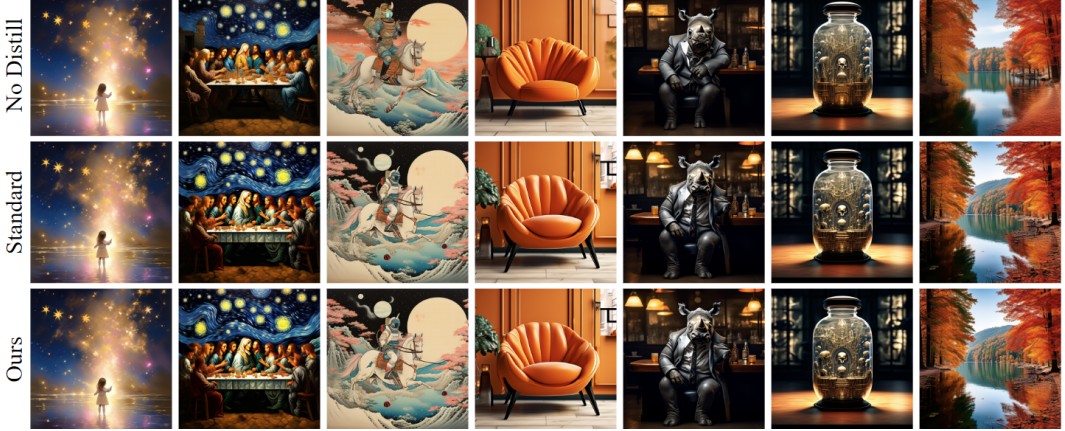

Figure 11: We show results for training with no distillation (top), with standard teacher distillation (middle), and then with our fast approximate teacher distillation (bottom). Ours gives the same good results as standard ablation, but with lower training cost.

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
