# OpenReview forum: "Accelerate High-Quality Diffusion Models with Inner Loop Feedback"
_ICLR.cc/2025/Conference — ICLR 2025 Conference Withdrawn Submission_

### Official Review · Reviewer_5KKk · 2024-10-30

**Soundness:** 3
**Presentation:** 2
**Contribution:** 3
**Rating:** 3
**Confidence:** 3

**Summary:**

This paper proposes a novel feedback mechanism for accelerating inference of the diffusion models. The method is based on the observation that the outputs from adjacent blocks and timesteps are similar. A feedback module is learned to skip blocks and timesteps. Learnable Feedback Rescaling and Fast Approximate Distillation are developed to improve training stability and efficiency. Both quantitative and quantitative results look promising.

**Strengths:**

1. The proposed method is novel and solid. It's interesting that the authors combines the view of inner blocks and timesteps, I think the scheduler would be a tricky part during training but it worked.
2. Experiment results look good and the acceleration is considerable.
3. Experiment design and discussion covers most of the topics.

**Weaknesses:**

1. My main concern is about writing, there are a lot to be improved, I just list some examples here:
- Avoid using verbal expressions and make the sentences concise. For example, “we already know that” (L96), “one does not need to store an entire additional set of models weights” (L92), “This is clearly not optimal” (L107), “feed its output features to the feedback” (module?) (L68), “for this caching” (L215) etc.
- Make the long sentences logically fluent. E.g., “Different diffusion models have different learned weight values, and training quickly, without overfitting, is quite challenging.” The first part of the sentence does not seem to be related to the second part.
- Always give a statement followed by an explanation. E.g., “One cannot train this feedback mechanism with basic random initialization; the magnitude of the feedback will be too large, and the training will diverge.” Why is the magnitude too large? If it’s experimental observation you should refer to the experiment section for why it fails and how to fix it.
- Be specific and always support with data.  E.g., “ILF is sometimes even better than the 20 step baseline”.

2. The motivation in L127 mentioned “redundancy” but not explained where it comes from. Is the motivation trying to improve caching since it has drawbacks (L107).

3. In the abstract “While many efforts to optimize diffusion models focus on achieving acceptable
image quality in extremely few steps (1-4 steps), our emphasis is on matching
best case results (typically achieved in 20 steps) while significantly reducing runtime.” Could you explain how “acceptable image quality” differs from “matching best case result”? Or is the emphasis that you are not using extremely few steps?

4. Is it possible to reduce the steps further? The design seems to be flexible to adjust the caching/skipping.

5. In fig. 8, why does image reward drop after a certain time for different feedback schedules?

**Questions:**

see weakness.

---

### Official Review · Reviewer_aqvz · 2024-11-01

**Soundness:** 3
**Presentation:** 2
**Contribution:** 2
**Rating:** 5
**Confidence:** 3

**Summary:**

This paper presents Inner Loop Feedback (ILF), a method to speed up diffusion model inference while preserving quality. ILF introduces a lightweight feedback module to predict future features in the denoising process, using outputs from selected model blocks. Through Fast Approximate Distillation and Learnable Feedback Rescaling, ILF achieves up to 1.8x faster generation times without sacrificing image quality. Results demonstrate ILF’s superior efficiency and quality compared to caching methods across various diffusion models and tasks.

**Strengths:**

- The paper presents a novel, efficient method for diffusion model acceleration, focusing on maintaining high-quality generation while reducing inference time.
- The approach is flexible, adaptable across multiple architectures (such as PixArt and DiT) and tasks.
- The ILF technique effectively balances speed and image quality, outperforming caching methods and achieving strong performance in human-aligned quality metrics.
- The Fast Approximate Distillation and Feedback-aware Inference Scheduling innovations contribute to efficient training and inference.

**Weaknesses:**

Table 1 shows that the performance improvement of ILF over the baseline method with steps=2 is not always significant.

In the Fast Approximate Distillation section, presenting the final noise expression with a formula would improve readability.

**Questions:**

1. In the Fast Approximate Distillation section, is it optimal to add noise to the teacher input at step t/2? Has there been any ablation study or analysis on the choice of t/2?
2. The derivation process of Equation (5) is not mentioned, which raises some questions about its meaning. Why does Equation (4) include the factor i×(n/m) multiplied by (t/1000)? And why is the maximum between the result and 10 taken—what is the rationale behind this?

---

### Official Review · Reviewer_1Y2K · 2024-11-02

**Soundness:** 3
**Presentation:** 2
**Contribution:** 1
**Rating:** 3
**Confidence:** 4

**Summary:**

This paper presents a new method for skipping inference steps in diffusion models. It does so by training an inner loop module that computes the activations of future steps directly from the current step. This module is trained using a self-distillation loss from a standard sampling, and requires a specialized noise scheduling. The proposed method is qualitatively and quantitatively compared to caching based approaches, and standard sampling.

**Strengths:**

- The paper presents a method for distilling denoising steps in a diffusion model using inner feature maps, which could encapsulate more information than standard distillation that does not observe inner feature maps.
- The qualitative results show no significant performance drop compared to the original model, while being much faster.
- Results are demonstrated on SoTA text-to-image models.

**Weaknesses:**

- **Comparison to Step Distillation Approaches:** ILF introduces a self-distillation approach to accelerate the inference of diffusion models. Since this concept has been explored in previous works [1,2,3], a comprehensive comparison with these methods is essential. However, ILF is primarily compared to a caching-based approach that does not involve step-skipping training. The only comparison to another distillation method is a brief qualitative ablation (Fig. 11) with just three samples. This raises a few concerns: (i) The experiment is poorly documented, and it is unclear which specific method was used. Although additional details are mentioned as being in the appendix, I could not find them. Could the authors please refer me to where these details are? (ii) As noted in lines 469-472, the results are reported to be similar to ILF.  This highlights the need for a more thorough comparison, including a numerical evaluation of performance and inference runtime of ILF against other step distillation methods.
In summary, I believe including some of these approaches ([1,2,3]) as baselines in the main results of the paper is essential.

- **Results of Less Denoising Steps Baseline:** When observing Tab. 1, one can notice that ILF offers a relatively minor improvement over a simple baseline which simply uses less denoising steps. This baseline performs similarly in both prompt-aware metrics and FID but does not require any additional data or model training, whereas ILF reportedly requires 100 H100 GPU hours (as mentioned in L343 and L480) making ILF a less favorable option. Furthermore, this baseline is excluded from the qualitative comparisons in Fig. 7. Could the authors please add these additional qualitative results to Fig. 7?

- **Inconsistency of Experimental Section:** The experimental section presented several variations of the proposed ILF. For instance, the main quantitative results (Tab. 1) use a different acceleration rate than the main qualitative comparisons (Fig. 7), and yet another rate is used for the ablation in Table 3. While I understand that these are all different choices for hyper-parameters, there should be some consistency, particularly between the main qualitative and quantitative results. Otherwise, it becomes very difficult to interpret the results presented.

- **Quality of Writing:** Some parts of the paper are poorly written, and hard to follow. The main issue for me is in the introduction section, which should be revised. Additionally, lines 350-360 and 469-472, both describing experimental settings, are specifically unclear.


[1] Liu, Xingchao, et al. "Instaflow: One step is enough for high-quality diffusion-based text-to-image generation." The Twelfth International Conference on Learning Representations. 2023.

[2] Sauer, Axel, et al. "Adversarial diffusion distillation." European Conference on Computer Vision. Springer, Cham, 2025.

[3] Meng, Chenlin, et al. "On distillation of guided diffusion models." Proceedings of the IEEE/CVF Conference on Computer Vision and Pattern Recognition. 2023.

**Questions:**

- As the method is non-trivial to implement, are the authors planning to release code for their approach?

---

### Official Review · Reviewer_Vpze · 2024-11-03

**Soundness:** 2
**Presentation:** 3
**Contribution:** 1
**Rating:** 1
**Confidence:** 4

**Summary:**

- This paper proposes to use a feedback sub-network that takes the feature maps of the previous diffusion step as an input, to then transform the feature maps of the next diffusion step for improving the inference without the need for distillation.

- The method is designed to add an additional block to an off-the-shelf transformer based diffusion denoiser, where this block takes the feature from block $b$ at step $t$, and predicts the features for the previous blocks with the previous diffusion step $t-r$.

- Instead of predicting the score $\epsilon(x_t, t)$ in the traditional way, this approach predicts $\epsilon(x_{t-r}, t-r)$

- For stabilizing the training they use learnable feedback scaling mechanism, where they multiple the feedback by scalar initialized to zero.

- The approach uses an additional loss term to train the model, specifically, they use the baseline model without the ILF as a teacher for predicting the noise when the image is less noisy, in particular at $\tilde{t}=t/2$.

**Strengths:**

- THe paper is well written.
- The method is examined both visually and numerically
- The method improves the quality of the generation while reducing the number of diffusion steps.
- The method is examined on 5 different diffusion models.
- The approach uses relatively light-weight network for the feedback.

**Weaknesses:**

- The method is examined on only transformer based networks, which limits the evaluation of the approach.
- The method is compared to caching that uses UNet architecture, making the comparison less effective.
- Line 264 there is a typo, please rephrase.
- The approach fine-tunes the network to predict score different than the one used in training, what is the advantage of that? also, how does it compare to fine-tuning the network itself without the feedback using this setup?
- The method is compared only to DPM-Solver++, and most methods benchmark ImageNet-64x64, which is not reported. Alternatively, you may report MS COCO-30k to align your evaluation with the literature.
- The method can be viewed as consistency model or even a self-distillation based approach, I was expecting to see comparison to SOTA methods from these classes.

**Questions:**

- Can this method be applied to UNet based diffusion as well?
- What are the motivation and advantages of predicting the previous epsilon estimation?
- How does this approach compare to consistency models?
- How does this method perform on ImageNet-64x64 or MS COCO 30K?
- How does it compare to approaches other than DPM-Solver++?

---

### Note · Authors · 2024-11-13

I have read and agree with the venue's withdrawal policy on behalf of myself and my co-authors.